# Epidemiological and phylogenetic analysis reveals *Flavobacteriaceae* as potential ancestral source of tigecycline resistance gene *tet*(X)

Rong Zhang [1,5], Ning Dong[2,5], Zhangqi Shen[3,5], Yu Zeng[1], Jiauyue Lu[1], Congcong Liu[1], Hongwei Zhou[1], Yanyan Hu[1], Qiaoling Sun[1], Qipeng Cheng[2,4], Lingbing Shu[1], Jiachang Cai[1], Edward Wai-Chi Chan[4], Gongxiang Chen [1✉] & Sheng Chen [2✉]

Emergence of tigecycline-resistance *tet*(X) gene orthologues rendered tigecycline ineffective as last-resort antibiotic. To understand the potential origin and transmission mechanisms of these genes, we survey the prevalence of *tet*(X) and its orthologues in 2997 clinical *E. coli* and *K. pneumoniae* isolates collected nationwide in China with results showing very low prevalence on these two types of strains, 0.32% and 0%, respectively. Further surveillance of *tet*(X) orthologues in 3692 different clinical Gram-negative bacterial strains collected during 1994–2019 in hospitals in Zhejiang province, China reveals 106 (2.7%) *tet*(X)-bearing strains with *Flavobacteriaceae* being the dominant (97/376, 25.8%) bacteria. In addition, *tet*(X)s are found to be predominantly located on the chromosomes of *Flavobacteriaceae* and share similar GC-content as *Flavobacteriaceae*. It also further evolves into different orthologues and transmits among different species. Data from this work suggest that *Flavobacteriaceae* could be the potential ancestral source of the tigecycline resistance gene *tet*(X).

[1] Department of Clinical Laboratory, Second Affiliated Hospital of Zhejiang University, School of Medicine, ZhejiangHangzhou, China. [2] Department of Infectious Diseases and Public Health, Jockey Club College of Veterinary Medicine and Life Sciences, City University of Hong Kong, Kowloon, Hong Kong, China. [3] Beijing Advanced Innovation Center for Food Nutrition and Human Health, College of Veterinary Medicine, China Agricultural University, Beijing, China. [4] State Key Lab of Chemical Biology and Drug Discovery, Department of Applied Biology and Chemical Technology, The Hong Kong Polytechnic University, Hung Hom, China. [5]These authors contributed equally: Rong Zhang, Ning Dong, Zhangqi Shen. ✉email: chengongxiang@zju.edu.cn; shechen@cityu.edu.hk

Excessive consumption of antimicrobials has resulted in rapid emergence of multidrug-resistant (MDR), extensively drug-resistant (XDR) and even pandrug-resistant (PDR) bacteria, which compromised the effectiveness of treatment of infectious diseases[1]. Carbapenem-resistant *Enterobacteriaceae*, *Acinetobacter baumannii* and *Pseudomonas aeruginosa*, the leading causes of nosocomial infections throughout the world, were listed as the critical priority group by WHO (https://www.who.int/medicines/publications/global-priority-list-antibiotic-resistant-bacteria/en/) in terms of urgency of need for alternative antibiotics[2]. Treatment of severe infections caused by these bacteria were typically restricted to the last-resort antibiotics including tigecycline and colistin[3]. In China, tigecycline, which was launched for clinical usage in 2010, is the primary choice for treatment of serious XDR bacterial infections. This drug was more preferred than colistin, another last-line antibiotic which was recently approved for clinical application at the end of 2017 but nevertheless exhibits toxicity[4]. With the emergence and global spread of the plasmid-borne colistin-resistance gene *mcr-1* in recent years, however, the clinical potential of colistin has been significantly compromised[5]. As a result, tigecycline is becoming increasingly important in treatment of infections caused by multidrug resistant organisms.

Tigecycline is a 9-t-butylglycylamido derivative of minocycline, which is the first drug of the glycylcycline class antibacterial agents[6]. It inhibits bacterial protein synthesis by reversibly binding to the 16S rRNA, hindering amino-acyl tRNA molecules from entering the A site of the ribosome and inhibiting elongation of peptide chains[7]. Chemical modification of tigecycline at the C-9 position of ring D led to enhanced binding to the target when compared to earlier classes of tetracyclines (tetracycline, doxycycline, and minocycline), and more effective evasion of common tetracycline resistance mechanisms[8]; tigecycline therefore exhibits broad-spectrum antimicrobial activity against MDR and XDR organisms. However, upon increasing clinical usage, tigecycline-resistant bacteria have emerged and posed a growing clinical concern[9]. The Tigecycline Evaluation and Surveillance Trial (TEST), which was a global antimicrobial susceptibility surveillance study, showed that between 2004 and 2013, tigecycline resistance rates in globally collected carbapenem-resistant *K. pneumoniae* (CRKP) and carbapenem-resistant *E.coli* (CREC) strains were 2.1% and 1.7%, respectively[10]. The MIC$_{90}$ of *A. baumannii*, for which no breakpoints were available for tigecycline previously, was 2 mg L$^{-1}$. 99% of *E. coli* strains and 91% of *K. pneumoniae* strains isolated from blood specimens were susceptible to tigecycline between 2012 and 2016, respectively[10]. The China Antimicrobial Surveillance Network (CHINET), which started monitoring tigecycline resistance in 2012, showed that the prevalence of tigecycline-resistant *Klebsiella* spp. slightly increased from 3.9% in 2012 to 5.4% in 2014, whereas the overall resistance rate of carbapenem-resistant *Klebsiella* spp. to tigecycline during this period was 16.8%. Data from the China CRE Network in 2015 showed that the rate of susceptibility of carbapenem-resistant *E. coli* to tigecycline was higher than that of CRKP (90.9% versus 40.2%)[11]. The rate of resistance of *Enterobacteriaceae* to tigecycline was 3.0% and 3.3% in 2015 and 2016, respectively[12,13].

Resistance to tigecycline is primarily due to over-expression of chromosome-encoding efflux pumps, mutations in the ribosomal binding sites and enzymatic inactivation[8,14]. Tet(X) is a flavin-dependent monooxygenase which is one of the most studied tigecycline-modifying enzymes[8,15,16]. Since identification of the *tet*(X) gene from the obligately anaerobic *Bacteroides* spp. in 2004, it has only been sporadically reported among strains of *Enterobacteriaceae*, *Comamonadaceae*, *Flavobacteriaceae* and *Moraxellaceae*[17,18]. To date, six orthologues of *tet*(X) have been reported, including *tet*(X) *per se* and *tet*(X1)~*tet*(X5), among which the plasmid-borne genes *tet*(X3), *tet*(X4), and *tet*(X5) were only reported recently and found to encode high-level tigecycline resistance phenotypes[18–20]. The emergence of plasmid-borne mobile tigecycline resistance genes recoverable from major pathogens including those of *Enterobacteriaceae* and *Acinetobacter* spp., poses serious threat to human health by compromising the antimicrobial efficacy of the last line drug tigecycline. Nevertheless, the origin, mechanism for transfer, and dissemination of the tigecycline resistance determinant *tet*(X), and its orthologues, remain poorly understood. In this study, we conduct a nationwide surveillance and genetic characterization of multiple genera of clinical isolates collected during the period 1994–2019 in China to fill this knowledge gap.

## Results

**Characterization of *tet*(X)-positive strains in clinical settings.** To determine the prevalence of tigecycline resistance and presence of *tet*(X)s in clinical *E. coli* and *K. pneumoniae* strains, we screened 1547 *K. pneumoniae* and 1250 *E. coli* isolates recovered from clinical samples that were collected from 77 hospitals located in 26 provinces in China during the period 1994–2019. Four out of the 1250 *E. coli* strains tested (0.32%) were resistant to tigecycline and found to carry the *tet*(X4) gene, whereas 62 (4.0%) of the 1547 *K. pneumoniae* strains tested were resistant to tigecycline, but the *tet*(X) gene could not be detected in these strains (Table 1). All the four *tet*(X4)-positive *E. coli* strains were isolated from hospitals in Zhejiang province (Fig. 1a). To assess the prevalence of tigecycline resistance and the presence of *tet*(X)s in other types of Gram-negative bacteria, we conducted epidemiological study in hospitals in Zhejiang Province. We screened 3892 clinical Gram-negative bacteria belonging to Bacteroidetes and Proteobacteria from different hospitals in Zhejiang Province. These bacteria include 2591 strains of *Acinetobacter* spp., 612 *S. maltophilia* strains, 376 Flavobacteriaceae strains, 136 strains of *Burkholderia*. spp., 108 strains of *Pseudomonas* spp. and 69 other Gram-negative isolates collected during the period 1994–2019 (Table 1). The rate of resistance to tigecycline varied among these strains, with *Pseudomonas* spp. exhibiting the highest rate (93/108, 86.11%), followed by *Burkholderia*. spp. (63/136, 46.32%), Flavobacteriaceae (95/376, 25.27%), *S. maltophilia* (43/612, 7.03%), and *Acinetobacter* spp. (103/2591, 3.98%). In addition, 68 Gram-negative isolates of other species were resistant to tigecycline (Table 1). These tigecycline-resistant bacteria were subjected to screening of the *tet*(X) orthologues, with results showing that a total of 102 (2.6%) bacterial strains carried the *tet*(X) orthologues. These include all 95 tigecycline-resistant Flavobacteriaceae strains, which were positive for *tet*(X2), one *P. xiamenensis* strain positive for *tet*(X2), one *A. nosomialis* strain positive for *tet*(X3), one *Citrobacter freundii* strain positive for *tet*(X4) and one *A. baumannii* strain positive for the *tet*(X5)-like gene, which exhibits 96.31% nucleotide identity and 95.0% amino acid identity to *tet*(X5)/Tet(X5) (Table 1, S1). These data indicated that all the tigecycline-resistant Flavobacteriaceae strains carried *tet*(X2), whereas the prevalence of *tet*(X)s in other Gram-negative bacteria is very low, and mostly belong to *tet*(X3-X5). The date at which the *tet*(X) orthologue was first detected in each bacterial species was summarized in Fig. 1b.

The 106 *tet*(X)-positive isolates were subjected to whole genome sequencing and determination of the type of antimicrobial resistance genes carried by these strains. Our data showed that each strain carried a number of antimicrobial resistance genes (ranging from 1 to 14), apart from *tet*(X), conferring multidrug resistance phenotypes. All the 95 *tet*(X2)-positive Flavobacteriaceae isolates were multidrug resistant and displayed high-level resistance to the last-line antibiotics colistin

**Table 1 Overview of isolation rate and tet(X) carriage rate of clinical isolates described in this study.**

| Classification of bacteria | No. of isolates | Year of isolation | Location (number of hospitals) | No. of Tigecycline resistant strains (percentage, %) | tet(X) Variants | | | | No.of tet(X)-carrying strains (%) |
|---|---|---|---|---|---|---|---|---|---|
| | | | | | tet(X2) | tet(X3) | tet(X4) | tet(X5) | |
| Enterobacterales | | | | | | | | | |
| *E. coli* | 1250 | 1998–2019 | 26 Provinces[a] (77) | 4 (0.32) | 0 | 0 | 4 | 0 | 4 (0.32) |
| *Klebsiella* spp. | 1547 | 1994–2019 | 26 Provinces (77) | 62 (4.00) | 0 | 0 | 0 | 0 | 0 |
| *Citrobacter* spp. | 14 | 2014–2019 | Zhejiang (1) | 0 | 0 | 0 | 1 | 0 | 1 (7.14) |
| *Enterobacter* spp. | 34 | 2014–2019 | Zhejiang (1) | 1 (2.94) | 0 | 0 | 0 | 0 | 0 |
| *Serratia* spp | 15 | 2014–2019 | Zhejiang (1) | 0 | 0 | 0 | 0 | 0 | 0 |
| *R. ornithinolytica* | 3 | 2014–2019 | Zhejiang (1) | 0 | 0 | 0 | 0 | 0 | 0 |
| Pseudomonadales | | | | | | | | | |
| *Acinetobacter* spp. | 2591 | 2004–2019 | Zhejiang (19), Henan (1) | 103 (3.98) | 0 | 1 | 0 | 1 | 2 (0.08) |
| *Pseudomonas* spp. | 108 | 2009–2019 | Zhejiang (1) | 93 (86.11) | 1 | 0 | 0 | 0 | 1 (100) |
| Burkholderiales | | | | | | | | | |
| *Burkholderia* spp | 136 | 2004–2013 | Zhejiang (4), Sichuan (1) | 63 (46.32) | 0 | 0 | 0 | 0 | 0 |
| Xanthomonadales | | | | | | | | | |
| *S. maltophilia* | 612 | 2004–2010 | Zhejiang (6), Beijing (3), Sichuan (2), Henan (2) | 43 (7.03) | 0 | 0 | 0 | 0 | 0 |
| Sphingobacteriales | | | | | | | | | |
| *S. mizuta* | 3 | 2004–2009 | Zhejiang (3) | 2 (66.67) | 3 | 0 | 0 | 0 | 3 (100) |
| Flavobacteriales | | | | | | | | | |
| *Elizabethkingia* spp. | 248 | 2004–2010 | Zhejiang (7) | 248 (100) | 46 | 0 | 0 | 0 | 46 (18.55) |
| *Chryseobacterium* spp. | 127 | 2004–2010 | Zhejiang (7) | 127 (100) | 48 | 0 | 0 | 0 | 48 (37.80) |
| *E. falsenii* | 1 | 2019 | Jilin (1) | 1 (100) | 1 | 0 | 0 | 0 | 1 (100) |
| Total | 6689 | 1994–2019 | 26 Provinces (77) | 747 (11.17) | 99 | 1 | 5 | 1 | 106 (1.58) |

*E. coli, Escherichia coli; R. ornithinolytica, Raoultella ornithinolytica; S. maltophilia, Stenotrophomonas maltophilia; S. mizuta, Sphingomonas mizutaii; E. falsenii, Empedobacte falsenii.*
[a]26 provinces included Anhui, Beijing, Fujian, Jiangsu, Shandong, Sichuan, Gansu, Guangdong, Guizhou, Zhejiang, Hainan, Hebei, Guangxi, Henan, Hunan, Hubei, Jilin, Jiangxi, Liaoning, Xinjiang, Tianjin, Shanghai, Shaanxi, Shanxi, Yunnan, and Chongqing.

(MIC ≥ 4 mg L$^{-1}$) and tigecycline (MIC ≥ 16 mg L$^{-1}$), as well as ceftzaidime, aztreonam, tobramycin. They were also highly resistant other antibiotics (Table 2, Supplementary Data 1). Notably, *Flavobacteriaceae* isolates were barely resistant to minocycline, with the resistance rate and MIC$_{90}$ being 1.05% and ≤1 mg L$^{-1}$, respectively. Similarly, the resistance rate for doxycycline was 32.63% (Table 2, Supplementary Data 1).

Analysis of the resistance gene profile of *Flavobacteriaceae* isolates showed that resistance to carbapenems in these organisms may be due to carriage of $bla_B$, $bla_{CME}$, $bla_{GOB}$, $bla_{IND-2b}$, $bla_{OXA-347}$, $bla_{IND-5}$, and $bla_{IND-8}$, resistance to macrolides (*ermF*), sulfonamides (*sul2*) and tetrcyclines (*tet*(36)) (Fig. 2). The *P. xiamenensis* isolate was found to carry the resistance genes *sul1, sul2, floR, cmx, aadA2,* and *aac(6')-Ib* and was resistant to ciprofloxacin, levofloxacin and trimethoprime/sulfamethoxazole. The two *tet*(X)-positive *Acinetobacter* spp. isolates (*A. baumannii* and *A.nosomialis*), despite carrying the *tet*(X) genes, were both sensitive to tigecycline and the clinically important antibiotics colistin and carbapenems. qRT-PCR analysis indicated that *tet*(X) gene in both strains were expressed with ΔΔCt values (average ± standard deviation) being 490 ± 19.343 and 12.339 ± 0.337, respectively. This might be due to mutations in these two genes that rendered them inactive. The detail mechanism should be further investigated. The four *E. coli* isolates and one *C. freundii* isolate shared highly similar antimicrobial resistance profiles, and exhibited high-level resistance to tigecycline (MIC = 4 and ≥8, respectively, Table 3). Conjugation results indicated that only the *tet*(X3) genes carried by two *E. coli* isolates and one *C. freundii* isolate could be successfully transferred to the *E. coli* strain EC600. All transconjugants exhibited 4–16-folds elevation in MIC when compared to the wild type recipients (Table 3). The result suggested that the *tet*(X) gene carried by strains of the *Flavobacteriaceae*, *S. mizutaii* and *Acinetobacter* spp. could not be readily transferred to *E. coli* under the test condition.

**Dissemination of tet(X)-positive strains.** Combined analysis of our data and that from literature enabled us to provide a comprehensive view on the current prevalence of *tet*(X) orthologues in different bacterial species. The *tet*(X) orthologous genes were detectable in two phyla, Bacteroidetes and Proteobacteria, including eight families, 16 genus and 28 species (Fig. 3, Supplementary Data 2)[21]. Bacteroidetes is composed of three large classes, Bacteroidia, Flavobacteria and Sphingobacteria[22]. The *tet*(X) gene is commonly distributed among strains of these classes, with the chromosomal-borne *tet*(X) orthologues, *tet*(X), *tet*(X1), and *tet*(X2) being the most dominant. Plasmid-borne *tet*(X) genes, *tet*(X3), and *tet*(X4), were also sporadically detected among members of Bacteroidetes, mostly in strains belonging to *Myroides* spp., *S. multivorum* and *E. brevis*. This study revealed the high prevalence of *tet*(X2) in *Flavobacteria*, in particular among species of *Chryseobacterium*, *Elizabethkingia* and *Empedobacter*, as well as in Sphingobacteria. It should be noted that detection of *tet*(X2) in clinical isolates of Bacteroidetes is mainly reported by this study (Fig. 3).

Proteobacteria isolates carrying *tet*(X) belonged to the two major clades of Betaproteobacteria and Gammaproteobacteria. Only two genera, *Delftia* and *Comamonas*, which belong to Betaproteobacteria, were previously reported to carry a chromosome-borne *tet*(X) genes (*tet*(X) and *tet*(X2)). Enterobacteriaceae, Pseudomonadaceae, Moraxellaceae and Morganellaceae are families of Gammaproteobacteria that carry *tet*(X). The dominant *tet*(X) orthologues reported among Gammaproteobacteria were mostly plasmid-borne genes (*tet*(X3)~*tet*(X5)), although *tet*(X1 and X2) were also reported in a few cases. Most of the previously reported *tet*(X) orthologues in Gammaproteobacteria were from animal sources, and this study has identified several *tet*(X) orthologues in clinical isolates of several different species (Fig. 3, Supplementary Data 2)[3].

**Analysis of origin and evolution trend of the TetX protein.** The *tet*(X) gene was first discovered in anaerobic *Bacteroides* spp.; yet it did not confer tigecycline resistance to this host strain since the TetX protein required oxygen to transform tigecycline[16]. *Bacteroides* spp. exhibited a median GC-content of 43.5%, whereas

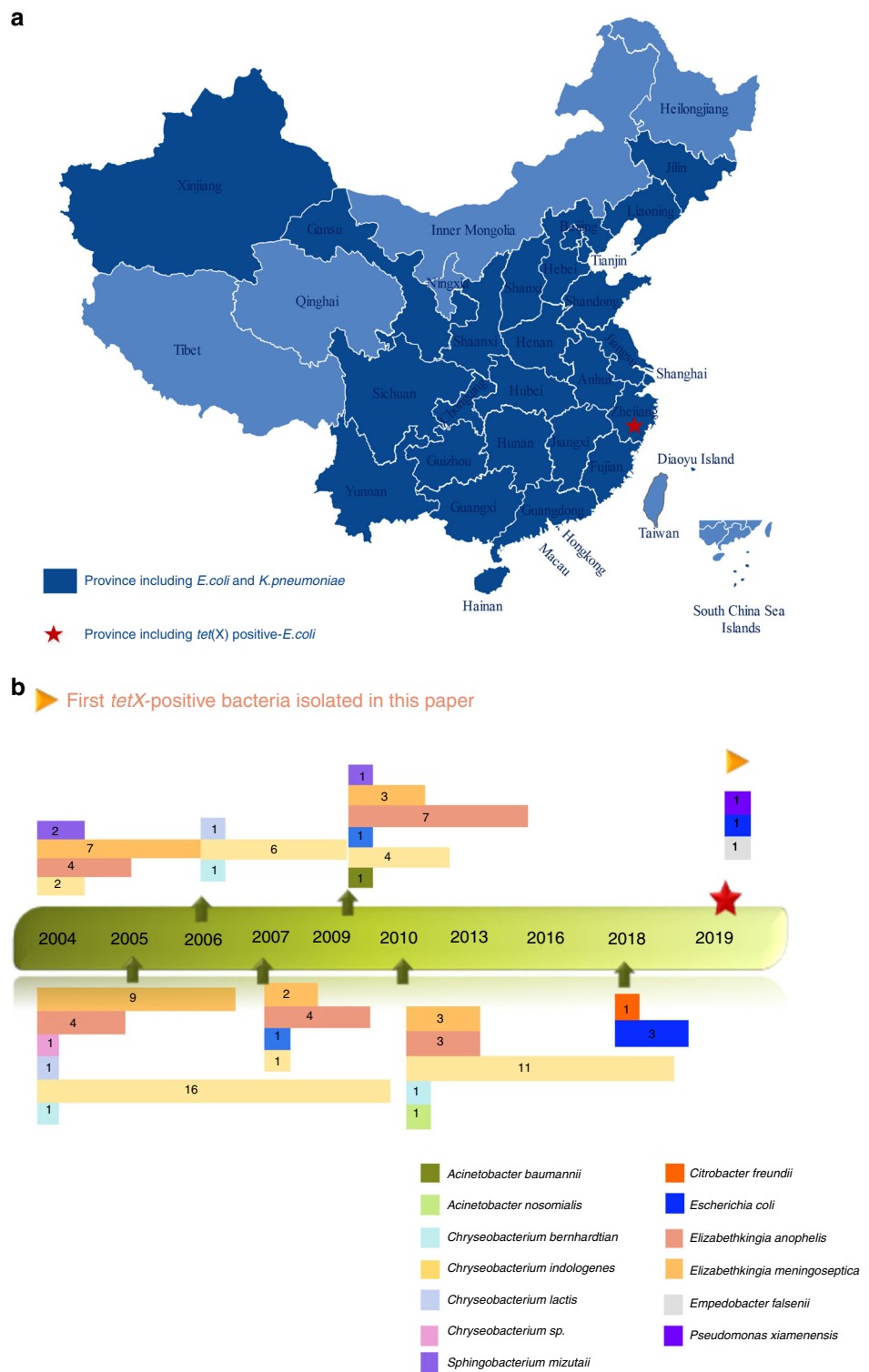

**Fig. 1 Distribution of *tet*(X)-positive bacterial strains. a** Distribution of *tet*(X)-positive clinical *E. coli* and *K. pneumoniae* strains in China. Dark blue background indicates provinces included in this surveillance; red star represents the province in which *tet*(X)-positive strains were isolated; light blue background indicates provinces in which no sample was collected. The map was created using Edraw Max v9.4. **b** Year of isolation of *tet*(X)-positive strains of specific bacterial species. The number of strains isolated each year is indicated.

that of the *tet*(X) gene was around 37%, suggesting that *tet*(X) did not originate from *Bacteroides* spp. The average chromosomal GC-content of *Acinetobacter*, *Enterobacteriaceae*, *Pseudomonas*, *Sphingobacterium*, *Flavobacteriaceae* were 39%, 50%, 61%, 40%, and 37% (except *Enpedobacter* which is around 31.6%), respectively. The similar GC-content of *Flavobacteriaceae* and the *tet*(X)

gene, the location of *tet*(X) genes on the chromosome among the *Flavobacteriaceae* strains, as well as the high carriage rate of *tet* (X2) by *Flavobacteriaceae* in this study, suggested that it could be an ancestral source of the *tet*(X) gene.

To validate our hypothesis and determine the phylogenetic profile of Tet(X), multiple sequence alignment of the Tet(X) was

**Table 2 Antimicrobial susceptibility profiles of non-Flavobacteriaceae tet(X)-positive isolates and the corresponding transconjugants.**

| Strain | Description | tet(X) Variants | MIC of antimicrobials (mg l⁻¹) | | | | | | | | | | | | | | | | |
|---|---|---|---|---|---|---|---|---|---|---|---|---|---|---|---|---|---|---|---|
| | | | TCC | TZP | CAZ | SFP | FEP | ATM | IPM | MEM | AMK | TM | CIP | LEV | DO | MNO | TGC | CS | SXT |
| EC600 | E. coli | - | 16 | ≤4 | 0.5 | ≤8 | ≤0.12 | ≤1 | ≤0.25 | ≤0.25 | ≤2 | ≤1 | ≤0.25 | 0.5 | 1 | 2 | ≤0.5 | ≤0.5 | ≤20 |
| SM2 | S. mizutaii | tet(X2) | ≥128 | 32 | ≥64 | ≥64 | ≥32 | ≥64 | 1 | 1 | ≥64 | ≥16 | ≥4 | ≥8 | 4 | ≤1 | ≥8 | ≥16 | 160 |
| AB1 | A. baumannii | tet(X5) | ≤8 | ≤4 | 8 | 32 | 16 | 16 | ≤0.25 | ≤0.25 | ≥64 | 8 | ≥4 | ≥8 | ≤0.5 | ≤1 | ≤0.5 | ≤0.5 | 80 |
| AN1 | nosomialis | tet(X3) | ≤8 | ≥128 | ≥64 | ≤8 | ≥32 | ≥64 | ≤0.25 | 0.5 | ≥64 | 4 | ≥4 | ≤0.12 | ≤0.5 | ≤1 | ≤0.5 | ≤0.5 | ≥320 |
| EC2 | E. coli | tet(X4) | ≤8 | ≤4 | 0.25 | ≤8 | ≤0.12 | ≤1 | 0.5 | ≤0.25 | ≤2 | 8 | ≤0.25 | 4 | ≥16 | ≥16 | 4 | ≤0.5 | ≥320 |
| CF1 | C. freundii | tet(X4) | ≤8 | ≤4 | 0.5 | ≤8 | ≤0.12 | ≤1 | 0.5 | ≤0.25 | ≤2 | ≤1 | ≤0.25 | ≤0.12 | ≥16 | ≥16 | ≥8 | ≤0.5 | ≤20 |
| J-CF1 | Transconjugant | tet(X4) | 16 | ≤4 | 0.25 | ≤8 | ≤0.12 | ≤1 | ≤0.25 | ≤0.25 | ≤2 | ≤1 | ≤0.25 | 0.5 | ≥16 | 8 | 2 | ≤0.5 | ≤20 |
| EC3 | E. coli | tet(X4) | 64 | ≤4 | 0.5 | ≤8 | ≤0.12 | ≤1 | ≤0.25 | ≤0.25 | ≤2 | 1 | 2 | 4 | ≥16 | ≥16 | 4 | ≤0.5 | ≥320 |
| J-EC3 | Transconjugant | tet(X4) | 16 | ≤4 | 4 | 16 | 1 | 16 | ≤0.25 | ≤0.25 | ≤2 | 4 | 4 | 1 | ≥16 | ≥16 | 2 | ≤0.5 | ≥320 |
| EC4 | E. coli | tet(X4) | ≥128 | ≤4 | ≥64 | 16 | 4 | ≥64 | ≥16 | ≥16 | ≥64 | ≥16 | 2 | 4 | ≥16 | ≥16 | 4 | ≤0.5 | ≥320 |
| J-EC4 | Transconjugant | tet(X4) | ≥128 | ≥128 | 32 | 16 | 2 | ≥64 | ≤0.25 | ≤0.25 | ≤2 | ≤1 | ≥4 | 2 | ≥16 | ≥16 | ≥8 | ≥16 | ≥320 |
| EF1 | E. falsenii | tet(X2) | ≤8 | ≥128 | 0.25 | ≤8 | ≤0.12 | ≤1 | ≥16 | ≥16 | ≤2 | ≤1 | ≤0.25 | ≥8 | ≤0.5 | ≤1 | ≥8 | 1 | ≤20 |
| PX1 | P. xiamenensis | tet(X2) | ≤8 | 16 | ≤0.12 | 16 | ≤0.12 | ≤1 | ≤0.25 | ≤0.25 | ≤2 | ≤1 | ≥4 | 1 | ≥16 | ≥16 | ≤0.5 | ≤0.5 | ≥320 |
| EC1 | E. coli | tet(X4) | ≤8 | ≤4 | ≤0.12 | ≤8 | ≤0.12 | ≤1 | ≤0.25 | ≤0.25 | ≤2 | ≤1 | ≤0.25 | 1 | ≥16 | ≥16 | 4 | ≤0.5 | ≥320 |

TCC ticarcillin/clavulanic acid, TZP piperacillin/tazobactam, CAZ ceftazidime, SFP cefoperazone/sulbactam, FEP cefepime, ATM aztreonam, IPM imipenem, MEM meropenem, AMK amikacin, TM tobramycin, CIP ciprofloxacin, LEV levofloxacin, DO doxycycline, MNO minocycline, TGC tigecycline, CS colistin, SXT trimethoprime/sulfamethoxazole.

performed. A total of 97 Tet(X) candidate proteins that returned hits with >60% identity with the *tet*(X) gene *per se* was retrieved from the NCBI database. Intriguingly, the reconstruction of a maximum phylogeny tree allowed us to identify two distinctive clades, one (clade I) containing a total of 28 Tet(X) candidate proteins carried by Flavobacteria and Sphingobacteria, among which the *Chryseobacterium* sp. that belongs to the *Flavobacteriaceae* family was the most dominant. This clade also contained other species from Flavobacteriales (*hymeobacter* sp., *Larkinella* sp.) and species that belong to Sphingobacteriales (*Pedobacter* sp., *Niabella* sp.). The other clade (clade II) contained the published TetX orthologues (Tet(X)~Tet(X5)) harbored by isolates that belong to diverse phylogenetic groups including Flavobacteriia, Sphingobacteria, Bacteroidales and Gammaproteobacteria (Fig. 4). The phylogenetic tree depicts a divergent evolutionary pattern of Tet(X). The first clade of Tet(X) protein evolved among Flavobacteriia and Sphingobacteria and did not spread to other species. Tet(X) candidate proteins in this clade were 60%~64% identical to that of Tet(X) in terms of amino acid sequence. The other clade evolved in a more divergent manner and covered all different variants of TetX proteins reported to date. Importantly, Tet(X)/Tet(X2) proteins in this clade were able to disseminate further into members of Bacteroidales and Gammaproteobacteria. Some of the Tet(X)/Tet(X2) proteins could further evolve into other variants of Tet(X) such as Tet(X3~5), especially in Gammaproteobacteria (Fig. 4). Interestingly, we detected two variants of Tet(X2) proteins among 97 strains of Flavobacteria and both belonged to clade II of Tet(X) proteins; this finding further indicates that Tet(X2) might be the origin of other TetX variants commonly detectable in clinical settings in China (Fig. 4).

**Genetic basis of transmission of the *tet*(X) genes**. Analysis of the *tet*(X) region in the *Flavobacteriaceae* isolates indicated that it could be located within mobilizable transposons, and carriage of multiple copies of *tet*(X2) by a single genome was detected in the genome of at least two *Flavobacteriaceae* isolates (*C. bernardetii* CB1 and *S. mizutaii* SM1). These findings indicated that *tet*(X) could undergo active gene transfer, especially in *Bacteroides sp.* isolates. The clade containing all published *tet*(X) orthologues comprises several separate sub-clades within the tree, suggesting the evolution routes of the *tet*(X) genes are highly diverse. The products generated through these evolution routes should be closely monitored.

To trace the transmission routes of *tet*(X), the gene environment of *tet*(X) in different organisms was analyzed. Genome mining indicated that the *tet*(X2) genes in completely sequenced *Flavobacteriaceae* strains were flanked by resistance genes *catB* and *ant(6)-I*, and were all located in integrative and conjugative elements (ICE) ranging from 54 to 97 Kbp in size (Fig. 5). ICE are modular mobile genetic elements integrated into a host genome and are passively propagated during chromosomal replication and cell division. Induction of ICE gene expression leads to excision and production of the conserved conjugation machinery (a type IV secretion system), which may promote genetic transfer to appropriate recipients[23]. The ICEs acted as a reservoir for the transmission of the *tet*(X) gene in *Flavobacteriaceae*. In *Acinetobacter* spp. and Enterobacteriaceae, *tet*(X3) and *tet*(X4) were typically associated with the genetic arrangements *xerD-tet*(X3)-*res-orf1*-IS*Vsa3* and IS*Vsa3-orf2-abh-tet*(X4), respectively[3]. Direct alignment of *tet*(X) genetic environments in *Flavobacteriaceae* with that of *Acinetobacter* spp. and *Enterobacteriaceae* returned no hits. However, a 397-bp fragment composed of partial fragments of the mobile element IS*Vsa3* and the *virD2* gene, which are uniquely detectable in members of *Acinetobacter*

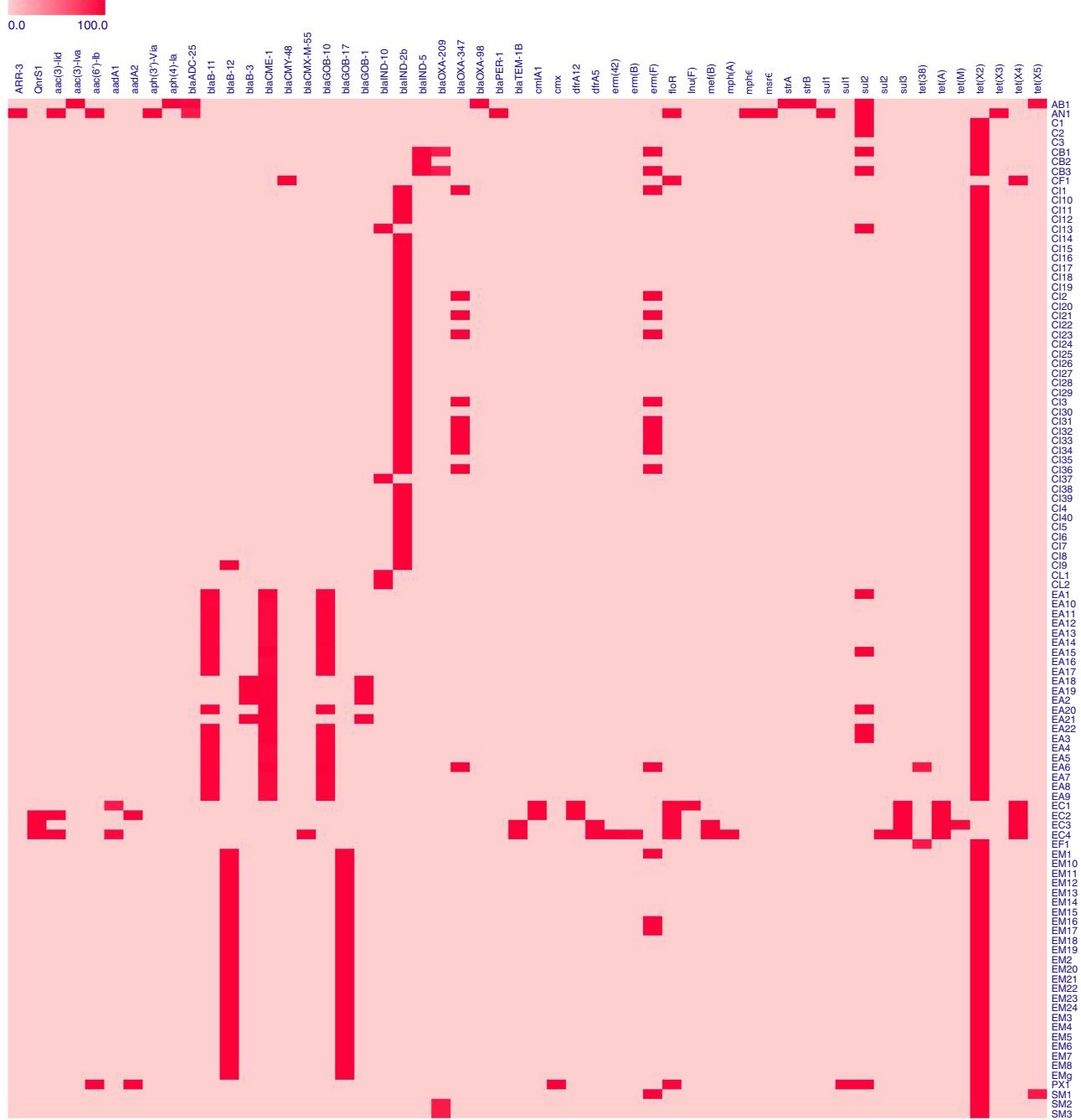

**Fig. 2 Heatmap of antimicrobial resistance genes in *tet*(X)-positive isolates in China.** The *X* axis represents the antimicrobial resistance gene carried by each strain. The *Y* axis indicates *tet*(X)-positive strains described in this study. Labels in the *Y* axis represent the species of the strain, AB *Acinetobacter baumannii*, AN *Acinetobacter nosomialis*, C *Chryseobacterium* sp., CB *Chryseobacterium bernardetii*, CF *Citrobacter freundii*, CI *Chryseobacterium indologenes*, CL *Chryseobacterium lactis*, EA *Elizabethkingia anophelis*, EC *Escherichia coli*, EF *Empedobacter falsenii*, EM *Elizabethkingia meningoseptica*, PX *Pseudomonas xiamenensis*, SM *Sphingobacterium mizutaii*. Red and pink colors indicate the presence and absence of the corresponding antimicrobial resistance genes, respectively.

spp. and *Enterobacteriaceae*, was identified in the genome of two *C. bernardetii* strains. Such findings suggested that genetic exchange among strains of *C. bernardetii* and *Acinetobacter* spp. /*Enterobacteriaceae* occurred, during which the *tet*(X) gene could have been exchanged among strains of different species during their evolution. Of note, such genetic exchange events could occur in processes other than conjugation. For example, some of the bacteria (e.g., *Acinetobacter* sp.) can be naturally competent and may have scavenged DNA fragments containing *tet*(X) from the environment.

To investigate the genetic basis of plasmid-borne *tet*(X) transmission, Nanopore and Illumina sequencing were performed to obtain the complete plasmid map. The *tet*(X4)-carrying plasmid in the *C. freundii* isolate belonged to IncN/X1type and was designated as pCF1. It was 51,531 bp in size and contained 70 ORFs, with a GC-content of 49.4%. This plasmid was found to be 99.87% identical to the 43,265 bp IncN plasmid pL2-43 (accession: KJ484641) from an *E. coli* isolate at 71% coverage (Supplementary Fig. 1). However, a 17 Kbp region bordered by an IS*26* element carrying the *tet*(X4) gene in pCF1 was absent in

**Table 3 Susceptibility of *tet*(X)-positive Flavobacteriaceae strains to commonly used antibiotics.**

| Antibiotic | MIC$_{50}$ (mg l$^{-1}$) | MIC$_{90}$ (mg l$^{-1}$) | Range (mg l$^{-1}$) | R% | I% | S% |
|---|---|---|---|---|---|---|
| TCC | ≥128 | ≥128 | ≤8–≥128 | 96.84% | 0.00% | 3.16% |
| TZP | ≥128 | ≥128 | ≤4–≥128 | 96.84% | 0.00% | 3.16% |
| CAZ | ≥64 | ≥64 | 32–≥64 | 100.00% | 0.00% | 0.00% |
| SFP | ≥64 | ≥64 | ≤8–≥64 | 71.58% | 17.89% | 10.53% |
| FEP | ≥32 | ≥32 | 2–≥32 | 94.74% | 4.21% | 1.05% |
| ATM | ≥64 | ≥64 | ≥64 | 100.00% | 0.00% | 0.00% |
| IPM | ≥16 | ≥16 | ≤0.25–≥16 | 96.84% | 0.00% | 3.16% |
| MEM | ≥16 | ≥16 | 1–≥16 | 96.84% | 0.00% | 3.16% |
| AMK | ≥64 | ≥64 | 16–≥64 | 98.95% | 0.00% | 1.05% |
| TM | ≥16 | ≥16 | ≥16 | 100.00% | 0.00% | 0.00% |
| CIP | ≥4 | ≥4 | 0.5–≥4 | 84.21% | 1.05% | 14.74% |
| LEV | ≥8 | ≥8 | 0.25–≥8 | 81.05% | 3.16% | 15.79% |
| DO | 8 | ≥16 | 1–≥16 | 32.63% | 26.32% | 41.05% |
| MNO | ≤1 | ≤1 | ≤1–≥16 | 1.05% | 0.00% | 98.95% |
| TGC | ≥8 | ≥8 | 4–>8 | 100.00% | Na | 0.00% |
| CS | ≥16 | ≥16 | ≥16 | 100.00% | Na | 0.00% |
| SXT | ≥320 | ≥320 | ≤20–≥320 | 77.89% | Na | 22.11% |

*TCC* ticarcillin/clavulanic acid, *TZP* piperacillin/tazobactam, *CAZ* ceftzaidime, *SFP* cefoperazone/sulbactam, *FEP* cefepime, *ATM* aztreonam, *IPM* imipenem, *MEM* meropenem, *AMK* amikacin, *TM* tobramycin, *CIP* ciprofloxacin, *LEV* levofloxacin, *DO* doxycycline, *MNO* minocycline, *TGC* tigecycline, *CS* colistin, *SXT* trimethoprime/sulfamethoxazole.
*R* resistant, *I* intermediate, *S* susceptible.
Na, not applicable since no intermediate value was defined for the corresponding antibiotic.

pL2-43, suggesting that the plasmid has undergone genetic recombination and was being transferred between different species of *Enterobacteriaceae* (Supplementary Fig. 1).

Complete sequence of the *tet*(X)-bearing plasmid recovered from four *E. coli* strains (EC1~EC4) was determined. The plasmid from *E. coli* strain EC1 was an IncX1 plasmid and was designated as pEC1-tetX4. It was 49,366 bp in size, contained 63 ORFs with a GC-content of 47.6%, and was 99.90% identical to the 57,104 bp plasmid pYY76-1-2 (accession: CP040929) recovered from an *E. coli* isolate, at 100% coverage. The plasmid carried multiple antimicrobial resistance genes including *tet*(X4), *tet*(A), *floR*, *aadA2,* and *lnu*(F). The *tet*(X4) gene was located in the genetic environment IS26-*abh*-*tet*(X4)-IS*Vsa3*, suggesting that it was acquired by horizontal gene transfer (Supplementary Fig. 2).

The plasmid from *E. coli* strain EC2 belonged to IncHI1B/HI1A/FIA and was designated as pEC2-tetX4. It was 193,021 bp in size and contained 228 ORFs, with a GC-content of 46.4%. It was 100% identical to the 190,128 bp plasmid pYPE10-190k-tetX4 (accession: CP041449), which was recovered from an *E. coli* isolate, at 99% coverage. The plasmid carried multiple antimicrobial resistance genes including *tet*(X4), *floR*, *aadA1*, *bla*$_{TEM-1B}$, and *qnrS1*. The *tet*(X4) gene was located in the genetic environment *orf1*-*abh*-*tet*(X4)-IS*Vsa3*-*orf2* (Supplementary Fig. 3).

Plasmid from *E. coli* strain EC3 and EC4 were identical, belonged to the IncFIB$_k$/FIA(HI1)/X1 type, and were designated as pEC3-tetX4 and pEC4-tetX4, respectively. They were 101,519 bp in length and contained 120 ORFs, with a GC-content of 49.8%. They were 99.7% identical to the 101,987 bp plasmid pYPE12-101k-tetX4 (accession: CP041443) recovered from an *E. coli* isolate, at 99% coverage. The plasmid carried multiple antimicrobial resistance genes including *tet*(X4), *mef*(B), *sul3*, *floR*, *tet*(A), *qnrS1*, *dfrA5*, *bla*$_{TEM-1B}$, and *tet*(M). The *tet*(X4) gene was located in the genetic environment IS*Vsa3*-*abh*-*tet*(X4)-IS*Vsa3*-*orf2* (Supplementary Fig. 4).

## Discussion

Tigecycline has become a viable alternative for treating severe infections, especially for those caused by XDR bacteria[8]. However, tigecycline-resistant bacterial strains have continued to emerge as a result of widespread antibiotic usage, and has become a major clinical concern[9]. Tet(X) is a tetracycline destructase,

which exhibits a unique enzymatic tetracycline inactivation mechanism[8,15,16]. A recent study reported the emergence of the plasmid-borne tigecycline resistance genes *tet*(X3)–*tet*(X5) in China, the products of which significantly compromised the treatment effectiveness of tigecycline[3,19,20]. A nationwide surveillance reported a high carriage rate of such genes among livestock[3]. The selective pressure imposed by the continuous application of tetracyclines in veterinary medicine could serve to maintain and spread the *tet*(X) genes among pathogenic microorganisms[19]. In clinical settings, dissemination of *tet*(X) orthologues clonally or via horizontal gene transfer could result in extensive colonization of tetracycline resistant organisms in human. In this study, we isolated *tet*(X) positive strains from hospitals located in the various provinces in China and conducted a comprehensive surveillance and molecular typing study.

According to data obtained in our surveillance, 11.17% of the test isolates exhibited tigecycline resistance, yet only 1.58% of all isolates carried *tet*(X)-like genes, suggesting that factors other than Tet(X) mediated the majority of tigecycline resistance in clinical settings. The over-expression of chromosomal efflux pumps of the RND family and those encoded by the plasmid-borne *tmexCD1-toprJ1* gene cluster, mutations in the ribosomal binding site such as the *rpsJ* gene, and mutations in the plasmid-mediated efflux pump genes *tet*(A) and *tet*(L) were reported to be associated with tigecycline resistance phenotypes[24–27]. Factors that confer the drug susceptibility phenotypes of tigecycline-resistant bacteria remained to be investigated.

We found that the majority of *tet*(X)-positive strains in clinical settings in China belonged to *Flavobacteriaceae*. Organisms in the family *Flavobacteriaceae*, which is within the phylum Bacteroidetes, were first isolated by Bergey[28], proposed by Jooste[29] and were included in the first edition of the *Bergey's Manual of Systematic Bacteriology*[22], in which the taxon was not formally described. The family includes the genus of *Chryseobacterium*, *Elizabethkingia*, *Wautersiella,* and 72 others as of 2019[30–35]. Species in the family have been isolated from various habitats, including freshwater, marine environments, soil, glacier and other environments, and are considered opportunist human pathogens[31,36,37]. Notably, *Elizabethkingia meningoseptica* and *Chryseobacterium indologene* are the most common clinical species of this family. These bacteria are phenotypically similar to

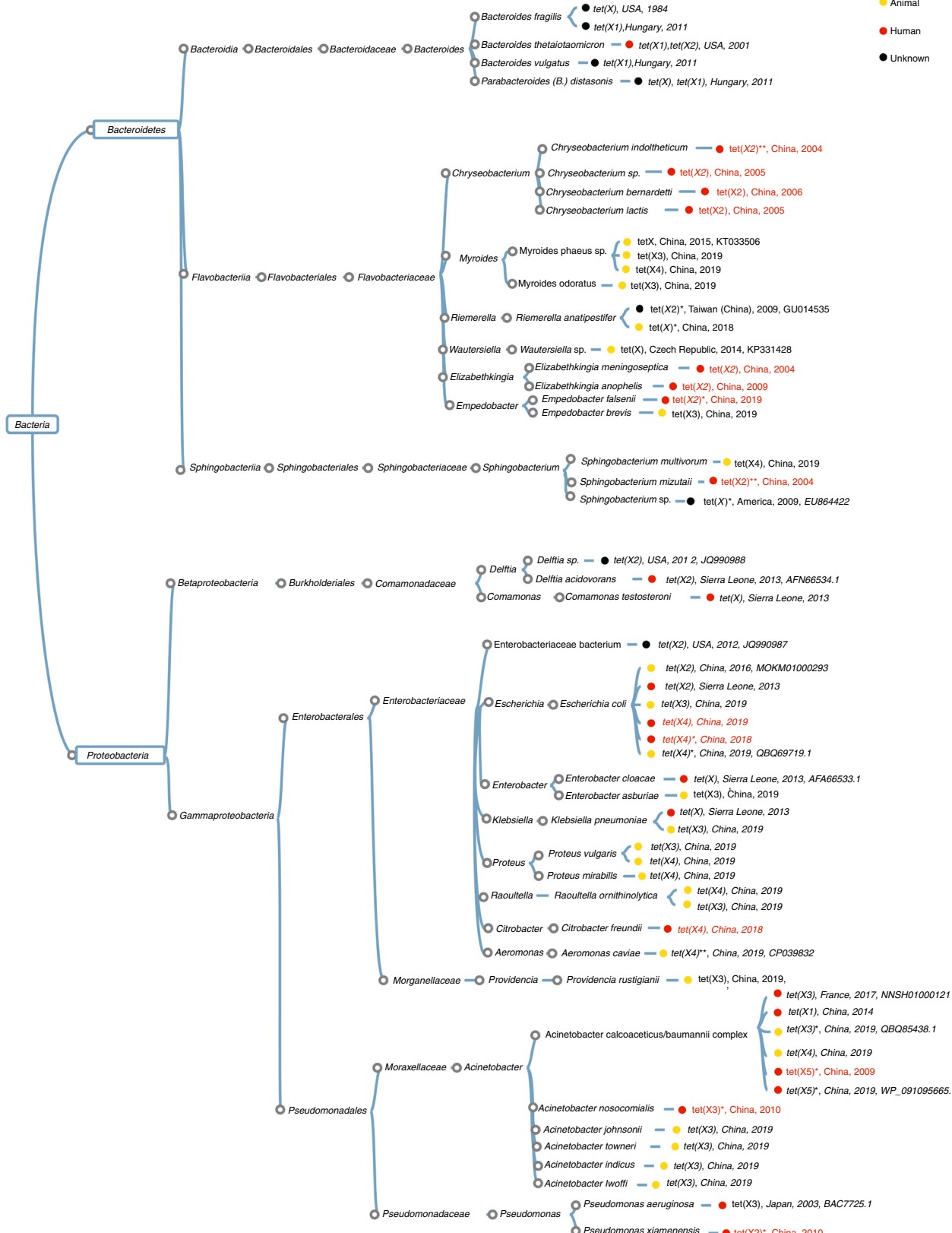

**Fig. 3 Taxonomy of Tet(X)-producing isolates analyzed in this and previous studies.** Asterisks (*) and (**) represent the location of the *tet*(X) gene in plasmid and chromosome, respectively. Text in red fonts depicts *tet*(X)-positive isolates tested in this study, and information of isolates with black fonts were retrieved from the literature (Supplementary Data 2). Strains isolated from different sources are shown by circles in different colors (human, red; animal, yellow; unknown, black). Year at the end of each branch denotes the year in which the first strain of the related taxonomy was isolated.

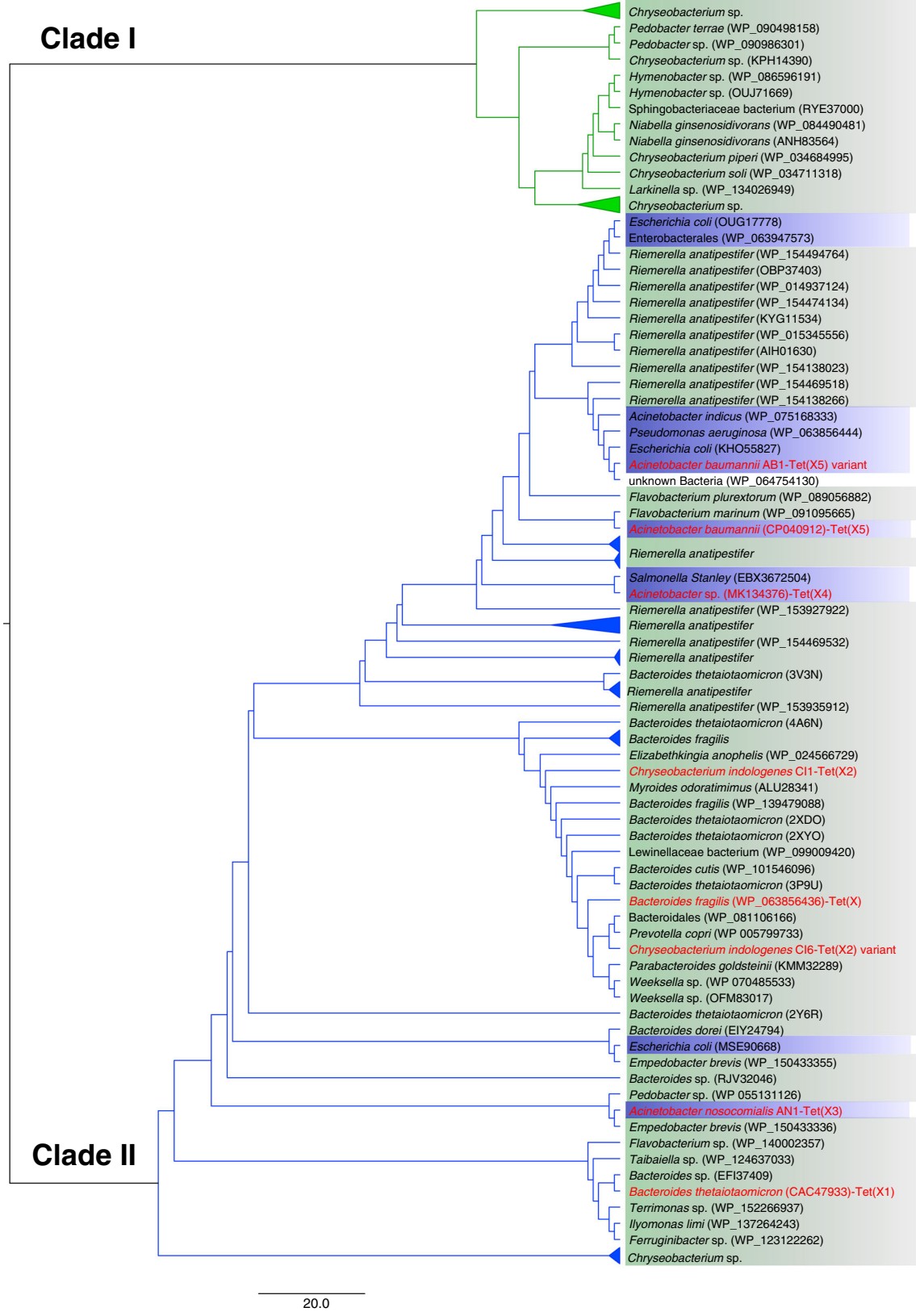

many of the Flavobacteriaceae and Flavobacteria-like organisms, as well as *S. mizutae*, which is not included in the family *Flavobacteriaceae*, rendering clinical identification difficult[38,39]. *E. meningoseptica* isolates may cause meningitis in premature newborns and immunocompromised patients[32,40]. In adults, various infection cases caused by these strains, which are often associated with a severe underlying illness such as pneumonia, endocarditis, postoperative bacteremia and meningitis, have been reported[40]. *C. indologenes* is associated with nosocomial infections and has been shown to cause a variety of invasive infections[41], such as primary bacteremia, catheter-related bacteremia, ocular infection, peritonitis, biliary tract infection and

**Fig. 4 Phylogeny of TetX generated by the maximum likelihood method.** Blue and green backgrounds denote bacteria that belong to Proteobacteria and Bacteroidetes, respectively. Published (with accession numbers) and representative *tet*(X) orthologues are labeled with red font. The evolutionary history is depicted by using the Maximum Likelihood method and on the basis of the JTT matrix-based model. The tree with the highest log likelihood (-3778.5180) is shown. Initial tree(s) for the heuristic search were obtained automatically by applying Neighbor-Join and BioNJ algorithms to a matrix of pairwise distances estimated using a JTT model, and then selecting the topology with superior log likelihood value. The tree is drawn to scale, with branch lengths measured in the number of substitutions per site. The analysis involved 97 amino acid sequences. All positions containing gaps and missing data were eliminated. There is a total of 131 positions in the final dataset. Evolutionary analyses were conducted with MEGA7 [2].

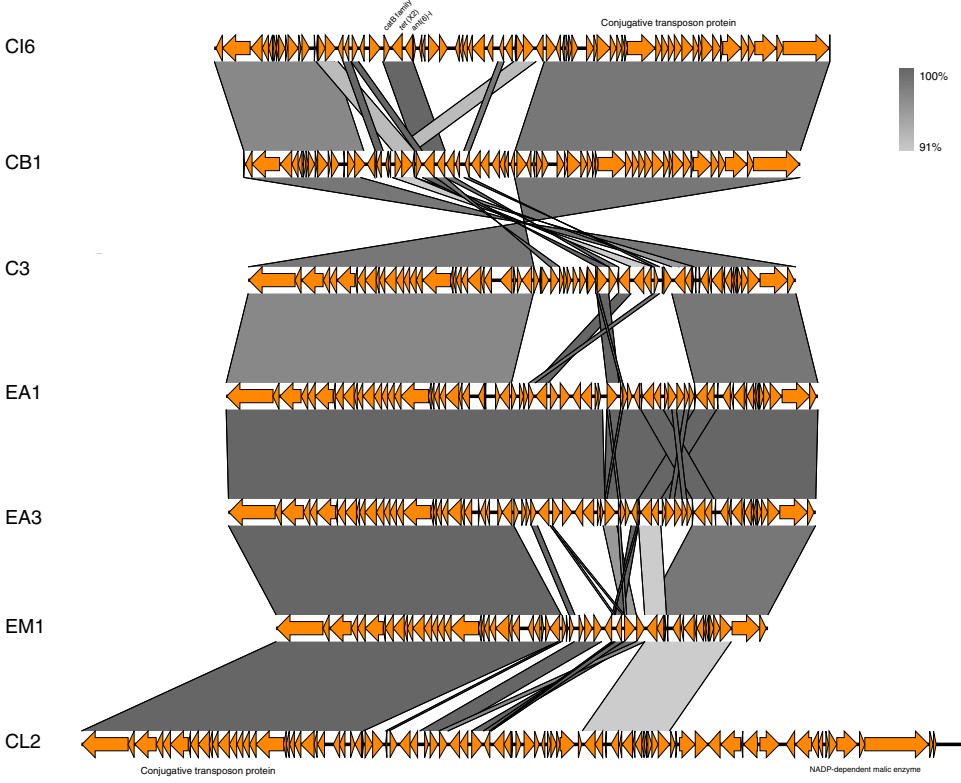

**Fig. 5 Alignment of integrative and conjugative elements with *tet*(X) in *Flavobacteriaceae*.** Yellow arrows indicate the ORFs. Shading area between different sequences indicate the aligned regions. Location of the *tet*(X) gene and the region responsible for conjugative transfer are labeled. ICE sequences aligned in this figure, from top to bottom, were from *tet*(X)-positive strains *Chryseobacterium indologenes* CI6, *Chryseobacterium bernardetii* CB1, *Chryseobacterium* sp. C3, *Elizabethkingia anophelis* EA1, *Elizabethkingia anophelis* EA3, *Elizabethkingia meningoseptica* EM1 and *Chryseobacterium lactis* CL2, respectively.

ventilator-associated pneumonia[42,43]. Interestingly, infections caused by *C. indologenes* have been reported mainly in Taiwan[42]. The genus *Sphingobacterium* can be occasionally isolated from blood, cerebrospinal fluid, urine and wounds. Strains of this genus may cause bacteremia, peritonitis and chronic respiratory infections[39]. To date, only three cellulitis cases caused by *Sphingobacterium* have been reported[39,44,45]. The source of infection was likely environmental in these three cases, suggesting that the bacterium is an opportunistic pathogen that may cause infection in immune-compromised patients.

Strains in the family of *Flavobacteriaceae* are usually resistant to multiple antibiotics prescribed for treating gram-negative bacterial infections, such as aminoglycosides, β-lactams, glycopeptides, quinolones or even carbapenems, the last line of defense against multi-drug resistant organisms, and thus constitute a major clinical concern.[32,41,46]. Infections caused by *Flavobacteriaceae* can be serious and are often associated with high death rate. The treatment outcome of infections caused by strains in this family depends on three factors: severity of the infection, immune status of the patient concerned, and the drug susceptibility phenotypes of the infecting agent[41,47].

The *tet*(X) gene was first identified in *Bacteroides fragilis* but it was found that the gene could not be expressed[48]. In 2009, Ghosh found a functional *tet*(X) gene in *Sphingobacterium* sp. strain, which was the first wild type bacterium of this species isolated[49]. According to result of a search of *tet*(X)-positive (including its variants) isolates in NCBI, the *tet*(X) genes have only been sporadically reported (Fig. 1). More recently, He et al. reported 77 *tet*(X3) or *tet*(X4) positive isolates from animals and human, including 51 *Enterobacteriaceae*, 16 *Acinetobacter*, four *Myroides spp.*, two *Raoultella ornithinolytica*, two *Empedobacter brevis*, and one each of *S. multivorum* and *Providencia rustigianii* strains[3]. There is significant difference in detection rate of *tet*(X3) or *tet*(X4) in organisms isolated from animals and humans (6.9%, 73 out of 1,060; 0.07%, 4 out of 5485). Only four *tet*(X4)-positive strains have been isolated from inpatients, including one *A. baumannii* (Jilin, 2013) and three *E. coli* strains (Zhejiang, 2016–17). We screened a total of 6692 isolates collected from hospitals that belonged to 13 different genera (Table 1) and showed that almost all *tet*(X)-positive species belong to *Flavobacteriaceae* (95, 25.27%). We found this gene in only 2 *Acinetobacter spp.* (0.08%), 6 *E. coli* (0.40%), 1 *B. diminuta* (100%), 4

*S. mizuta* (100%) and 1 *P. xiamenensis* (100%) strain. Notably, we did not find this gene in strains of *K. pneumoniae* and other common *Enterobacteriaceae* species.

Aerobic culture of *Flavobacteriaceae* is known to degrade tigecycline and exhibit resistance to this drug. The chromosomal GC-content of *Flavobacteriaceae* strains (37%) was similar to that of the *tet*(X) gene. Phylogenetic analysis revealed a divergent evolutionary pattern of *tet*(X), among which the route involving *Flavobacteriaceae* generated a major clade, suggesting that it can be regarded as an ancestral source of this gene. This hypothesis is substantiated by the genetic organization and synteny of genes upstream and downstream of *tet*(X) in *Flavobacteriaceae*. The *tet*(X) gene is located on ICE elements, but conjugal transfer of *tet*(X) from *Flavobacteriaceae* to other bacteria using *E. coli* recipient was not demonstrated, suggesting that strains of *Flavobacteriaceae* may lack other gene functions necessary for transfer of *tet*(X) under our test condition. The recovery of a 397-bp *Acinetobacter/Enterobacteriaceae*-restricted DNA fragment in *Flavobacteriaceae* indicates that genetic exchange can occur among members of the phylum Proteobacteria and Bacteriodetes, and that various strains in the family of Enterobacteriaceae may acquire the *tet*(X) genes during this process. Despite these observations, we cannot rule out the possibility that there was a non-*Flavobacteriaceae* ancestor which passed the ancestral *tet*(X) gene to *Flavobacteriaceae* and Proteobacteria on occasions such as polyphyletic events.

## Methods

**Retrospective screening of *tet*(X)-carrying clinical isolates**. The study comprised two phases, both of which involved analysis of clinical isolates collected from hospitals in China. The first focused on surveillance of *tet*(X) genes in clinical *E. coli* and *K. pneumoniae* isolates collected from 77 hospitals located in 26 provinces and municipalities in China during the period 1994 to 2019, including Anhui, Beijing, Fujian, Gansu, Guangdong, Guangxi, Guizhou, Hainan, Hebei, Henan, Hubei, Hunan, Jilin, Jiangxi, Liaoning, Jiangsu, Shandong, Shanxi, Shaanxi, Shanghai, Sichuan, Tianjing, Xinjiang, Zhejiang, Yunnan, and Chongqing. The second phase included surveillance of *tet*(X) genes in non-duplicated clinical strains of different Gram-positive bacterial species including Flavobacteriaceae, *Acinetobacter*. spp., *Burkholderia*. spp., *Stenotrophomonas maltophilia, P. aeruginosa* and other Gram-negative bacteria collected from seven hospitals located in Zhejiang Province. All isolates were collected as part of an active surveillance process conducted in China. Ethical permission for this study was given by the Zhejiang University ethics committee with the reference No. 2019-074. All strains were subjected to species confirmation using 16S rRNA gene-based sequencing and matrix-assisted laser desorption ionization/time of flight mass spectrometry (MALDI-TOF MS) (Bruker Daltonik GmbH, Bremen, Germany). All isolates were subjected to screening of *tet*(X) orthologues by PCR and Sanger sequencing according to methods published previously[3]. All primers used in this study are listed in Supplementary Table 1.

**Antimicrobial susceptibility tests and transconjugation**. The minimum inhibitory concentrations (MICs) of *tet*(X)-positive strains against 15 commonly used antibiotics (imipenem, meropenem, ceftzaidime, cefepime, ciprofloxacin, levofloxacin, doxycycline, minocycline, tigecycline, tobramycin, colistin, ticarcillin/clavulanic acid, piperacillin/tazobactam, cefoperazone/sulbactam, trimethoprim/sulfamethoxazole) were determined by the VITEK 2 COMPACT automatic microbiology analyzer, and interpreted according to the Clinical and Laboratory Standards Institute guidelines, except tigecycline and colistin[50], the breakpoints of which were interpreted according to the European Committee on Antimicrobial Susceptibility Testing (EUCAST)[51].

The efficiency of plasmid conjugation between 27 *tet*(X)-carrying strains and the *E.coli* strain EC600 was tested using the mixed broth method[52]. These strains included 17 randomly selected Flavobacteriaceae strains (5 *C. indologenes*, 3 *E. meningoseptica*, 3 *E. anophelis*, 2 *C. bernardetii*, 1 *C. lactis*, 2 *Chryseobacterium* sp., 1 *E. falsenii*), as well as strains of *E. coli* (3), *C. freundii* (1), *S. mizuta* (3), *Acinetobacter* spp. (2), *A.nosomialis* (1), *A. baumannii* (1), and *P. xiamenensis* (1). Transconjugants were selected on LB agar plates supplemented with 1 µg mL$^{-1}$ tigecycline and 600 µg ml$^{-1}$ rifampicin. PCR was conducted to confirm the presence of *tet*(X) genes; MALDI-TOF MS was applied to confirm the species identity of the transconjugants. MIC of the transconjugants was tested using the aforementioned method.

**RNA isolation and quantitative real-time PCR**. Quantitative real-time PCR (qRT-PCR) was conducted to test whether *tet*(X) genes in the two *Acinetobacter* spp. strains susceptible to tigecycline were expressed. Briefly, total RNA was extracted using the QIAGEN RNeasy Mini Kit (Qiagen Inc., Valencia, CA, USA), following the manufacturer's instructions. Extracted RNA was treated with Invitrogen TURBO DNA-free Kit (Ambion, Austin, TX, USA), followed by reverse transcription using the Invitrogen SuperScript™ III Reverse Transcriptase (Invitrogen, Carlsbad, CA, USA). qRT-PCR was conducted in a Light Cycler 480 (Roche Diagnostics), using primers in Supplementary Table 1[53]. All reactions were performed in triplicate. The 16S rRNA gene of *Acinetobacter* spp. were amplified with previously published primers and used as endogenous reference genes[54]. *A. baumannii* ATCC 19606 with a tigecycline MIC of 0.5 µg mL$^{-1}$ was used as the reference strain. Relative expression levels of the *tet*(X) genes were obtained by the ΔΔCT analysis method.

**Whole-genome sequencing and bioinformatics analysis**. Genomic DNA was extracted from overnight cultures by using the PureLink Genomic DNA Mini Kit (Invitrogen, Carlsbad, CA, USA). All isolates carrying the *tet*(X) genes were subjected to whole genome sequencing using the HiSeq platform (Illumina, San Diego, CA). Genome assembly was conducted with SPAdes v3.12.0[55]. Oxford nanopore MinION sequencing was conducted to obtain the complete genome sequences of strains carrying the *tet*(X) gene, using the SQK-RBK004 sequencing kit and flowcell R9.5[56]. Hybrid assembly of Illumina and nanopore sequencing reads was constructed using Unicycler v 0.3.0[57]. Genome sequences were annotated using RAST v2.0 with manual editing[58]. Multilocus sequence typing (MLST) of all *Enterobacteriaceae* isolates was conducted using MLST v2.11[59]. Acquired antibiotic resistance genes were identified by ResFinder 2.1[60]. Heatmap of antimicrobial resistance genes was plotted using Genesis v1.7.7[61]. Plasmid replicons were analyzed using PlasmidFinder 2.1[62]. Insertion sequences (ISs) were identified using ISfinder v2.0[63]. The genetic location of the *tet*(X) gene was determined by aligning the contigs carrying *tet*(X) with complete genome sequences in the NCBI database and those generated in this study. 16S rRNA genes of all isolates were obtained using Barrnap v0.9[64]. Integrative and conjugative elements were predicted using ICEberg2[65]. GC-content of all genome assemblies were calculated using CLC Genomics Workbench v9 (Qiagen, CA, USA).

**Prevalence of *tet*(X) orthologues in Gram-negative bacteria**. We searched PubMed with no language restriction for articles that contained the terms *tet*(X), *tet*(X) and tigecycline resistance, tigecycline resistance gene to retrieve information on previously published *tet*(X)-positive isolates, including isolation source, year of isolation and location of the *tet*(X) genes. These isolates were classified according to Bergey's Manual of Systematic Bacteriology[35]. Information of *tet*(X)-positive strains both from previous studies and this study were plotted to create a classification figure using Adobe Illustrator v22.1.

**Sequence acquisition and alignment of TetX**. To identify sequence homologous to *tet*(X) and its orthologues, a BLASTp v.2.10.0 search was performed using the amino acid sequences of TetX to TetX5 as query sequences. In order to avoid BLAST hits from very closely related species, uncultured environmental samples were excluded from the search and the max target sequences acquired were 100. The top unique protein sequences were selected and submitted to the Guidance2 server to evaluate the quality of the alignment and identify potential regions and sequences that reduce the quality of alignment[66]. Multiple sequence alignment was performed using MUSCLE v3.8.31 with default parameters[67]. A model with the least score (JTT + I) evaluated using the Models function in MEGA 7 was selected as the best amino acid substitution model to reconstruct the Maximum-Likelihood tree. Gaps and missing data were treated as partial deletions. Results were validated using 1000 bootstrap replicates[68].

**Reporting summary**. Further information on research design is available in the Nature Research Reporting Summary linked to this article.

## Data availability

The whole genome sequencing data of all strains in this study have been submitted to the NCBI database under the BioProject accession number PRJNA595705. The 16S rRNA gene sequences of all strains in this study have been deposited in the NCBI GenBank database under accession numbers MT793124-MT793229.

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

## Acknowledgements

This study was supported by the National Natural Science Foundation of China (81861138052, 81871705, 31761133004, and 81772250).

## Author contributions

R.Z., N.D., Z.S., G.C., and S.C. designed the study. R.Z., Y.Z., J.L., C.L., H.Z., Y.H., Q.S., Q.C., L.S., and J.C. collected the strains, conducted the screening experiment, bacteria characterization and literature review. S.C., R.Z., N.D., E.W.C.C., and G.C. analyzed and interpreted the data and wrote the manuscript. All authors reviewed, revised, and approved the final report.

## Competing interests

The authors declare no competing interests.
