## [Peer Review File · Nature Communications]

Reviewers' Comments:

Reviewer #1:

Remarks to the Author:

This is a descriptive study whereby the authors conducted a widespread survey (22 provinces in China) of clinical bacterial isolates (restricted to *E. coli* and *K. pneumoniae*), finding very few were both positive for resistance to tigecycline and positive for a tet(X) orthologue. Next, they analyzed a large collection of Gram-negative clinical strains (n=3,692) from several hospitals in one province and found that 25.8% of strains from the family Flavobacteriaceae were positive for tet(X2). The %GC content, the predominantly chromosomal location, and the relatively high prevalence in these species suggests that Flavobacteriaceae are the ancestral source of tet(X) orthologues.

The title is not informative

The abstract could better reflect the reasoning underlying the main conclusion.

There are numerous grammar, syntax and stylistic errors in the text (the first sentence of the abstract is but one example).

The results could be condensed by reducing the replication of results shown in the tables. The paragraphs describing the sequenced plasmids (Ln 252-263) should be condensed.

Ln 161 – Two *Acinetobacter* were positive for tet(X) but negative for a phenotype. Were the genes expressed (a simply RT-PCR experiment could address this question)?

Ln 207. This reviewer concurs with the conclusion about ancestry, although the authors should point out in the discussion that there could be a non-Flavobacteriaceae ancestor that passed the ancestral tet(X) gene to this family and Proteobacteria (i.e., polyphyletic events).

Figure 4 – authors need to label the clades.

L305 – as with the introduction, the authors attribute the emergence of tigecycline-resistant bacteria to “abusive use of antibiotics.” They made no attempt to quantify how antibiotics were being used. In fact, “any” use of tetracycline antibiotics has the potential to select for such resistance, whether anyone considers this abusive or not. Best to avoid value judgements in the absence of data.

L311 – the authors argue that the “molecular mechanisms underlying dissemination of these genes in clinical settings” is poorly understood...and presumably, this is the rationale for their study (next sentence). Their study, however, has nothing to do with dissemination in clinical settings. They are able to provide a prevalence value based on a taxa, and they make a reasonable argument that Flavobacteriaceae is the ancestral source of tet(X) genes. They were unable to demonstrate horizontal gene exchange, and the fact that none of the non-Flavobacteriaceae examined had a tet(X2) orthologue suggests that horizontal gene exchange, if it occurs, is infrequent. In this case, dissemination in clinical settings is really an issue of infectious organisms colonizing new human hosts – the language should reflect greater precision of meaning.

L263-266 – the authors speculate that genetic exchange occurred sometime in the past. This argument is consistent with the limited data available (previous sentence). I recommend that the authors point out that this did not have to occur by conjugation. Some of the bacteria in question (e.g., *Acinetobacter*) can be naturally competent and may have scavenged the DNA in the environment (both are considered environmental bacteria). It would be useful if the authors could

estimate when such an exchange might have occurred (millions of years ago?).

The supplement, as provided to the reviewer, was unreadable (excel page spread across multiple pdf pages), but presumably this is where the reader can figure out which strain codes apply to which strains. In places where the strains can be given more complete descriptions (e.g., genus and species), it is best to provide this information for the convenience of your reader (e.g., Figure 5).

Figure 1A is labeled as describing the "prevalence" but it actually describes the "distribution"

Ln 383 – authors need to point out that these are all clinical isolates in the methods – and identify their human subjects clearance (or reason for exemption).

Ln463 – whole genome sequences are deposited in GenBank, but it looks like there were other 16S sequences that also need to be deposited, correct?

Reviewer #2:

Remarks to the Author:

I would like to thank the authors for their manuscript. Resistance to novel antibiotics such as tigecycline is an important topic and the molecular epidemiology of the tet(X) this study adds to this area. However, I have questions, suggestions and thoughts to add to the manuscript. Please refer to the points below for a complete review:

Abstract

1. Line 30: replace report with resort

Introduction:

2. Line 104: The authors could add specifically that they are looking at multiple genera of clinical isolates specifically as there seems to be a knowledge gap in this area.

Results

3. Line 110-113: While the sample size is large and covered many hospitals in a large area, I wanted to confirm with the authors if they can assume that their study population can be representative of China. Were samples collected as part of an active surveillance process? If they could elaborate a little more on this in the manuscript (either in methods or in their results section), that would be helpful.

4. Line 122: Replace Burkholderi with Burkholderia. Please check the manuscript for additional changes to Burkholderia spp.

5. Line 196: Please provide the reference for the study that looked at tet(X) from animal sources that was used to create Figure 3.

6. Figure 3: "Text with red fonts depicts tet(X)-positive isolates". Please expand, do you mean these isolates are those that were tested in this current study?

7. Lines 223-225: "The first clade of Tet(X) protein evolved among Flavobacteriia and Sphingobacteria and did not spread to other species. All Tet(X) proteins in this clade remained in the Tet(X) and

Tet(X2) family". It would be helpful if the authors could add which isolate had which Tet(X) gene variant in Figure 4. Just by looking at Figure 4, I am unable to tell that the isolates in the first clade had Tet(X) and Tet(X2).

8. Figure 4: I do not understand the coding scheme of Figure 4. The authors mention "Orange, blue, yellow and green backgrounds denote bacteria belonging to Bacteroidia, Flavobacteria, Gammaproteobacteria and Sphingobacteria respectively". However, the backgrounds are discrepant. For instance, the authors have highlighted Salmonella Stanley (which should be classified as Gammaproteobacteria) as blue which indicates that it is belonging to Flavobacteria. Unless I am mistaken about the purpose of the blue/green highlights in the figure, please cross-check and/or re-write.

Discussion:

9. The authors could also include the importance of finding tet(X) genes in both clinical and human isolates and what it means for human health due to horizontal gene transfer.

10. Another important discussion point that the authors should consider adding in their discussion is while they observed high levels of tigecycline resistance, few isolates carried tet(X) genes. They should consider highlighting other genes that are responsible for conferring tigecycline resistance.

11. The authors should also consider why the tet(X) gene may be important for other tetracycline antibiotics, especially due to their widespread use in veterinary medicine.

Overall thoughts:

The manuscript is well-written and has important implications on the molecular epidemiology, evolution and transmissibility of the tet(X) gene. The authors articulated clearly why tigecycline resistance is a major problem and highlighted the missing gaps in information related to surveillance of the tet(X) gene in China. I also appreciate the inclusion of already existing tigecycline resistance surveillance system in the introduction section which gives the readers a good idea about what data is being collected and what is lacking. There were a few points in the results section where I was a little unclear about the results. If the authors could address these concerns, this would further strengthen the results. While going through Table 1, I did notice that there may be phenotypic and genotypic discrepancies between phenotypic tigecycline resistance and presence/absence of tet(X) gene. The authors should consider this point as well. The Materials and Methods section had information to ensure reproducibility of the work. I would recommend the authors to consider adding additional discussion points in their manuscript highlighted above to strengthen their conclusions.

Overall, I would recommend minor revisions to this manuscript before being accepted for publication.

Thank you very much,

Sanjana Mukherjee

Response to reviewers' comments

Reviewer #1 (Remarks to the Author):

This is a descriptive study whereby the authors conducted a widespread survey (22 provinces in China) of clinical bacterial isolates (restricted to *E. coli* and *K. pneumoniae*), finding very few were both positive for resistance to tigecycline and positive for a tet(X) orthologue. Next, they analyzed a large collection of Gram-negative clinical strains (n=3,692) from several hospitals in one province and found that 25.8% of strains from the family Flavobacteriaceae were positive for tet(X2). The %GC content, the predominantly chromosomal location, and the relatively high prevalence in these species suggests that Flavobacteriaceae are the ancestral source of tet(X) orthologues.

Q1: The title is not informative

Response: We have modified the title to “Epidemiological and phylogenetic analysis revealed Flavobacteriaceae as potential ancestral source of tigecycline resistance gene *tet(X)*”.

Q2: The abstract could better reflect the reasoning underlying the main conclusion.

Response: We have modified the abstract and included evidences supporting our conclusions. The following sentence was added to the abstract: “with Flavobacteriaceae being the dominant (97/376, 25.8%) bacteria. In addition, tet(X)s were found to be predominantly located on the chromosomes of Flavobacteriaceae and shared similar GC-content as Flavobacteriaceae. It could also be further evolved into different orthologues and transmitted among different species. The data from this work suggest that Flavobacteriaceae could be the potential ancestral source of the tigecycline resistance gene *tet(X)*.”

Q3: There are numerous grammar, syntax and stylistic errors in the text (the first sentence of the abstract is but one example).

Response: The manuscript has been edited by a native English speaker and all grammatical, syntax and stylistic errors should have been removed.

Q4: The results could be condensed by reducing the replication of results shown in the tables. The paragraphs describing the sequenced plasmids (Ln 252-263) should be condensed.

Response: We have tried our best to condense the results section. Lines 252-263 have also been condensed as follows “The ICEs acted as a reservoir for the transmission of the tet(X) gene in Flavobacteriaceae. In *Acinetobacter* spp. and Enterobacteriaceae, tet(X3) and tet(X4) were typically associated with the genetic arrangements *xerD-tet(X3)-res-orf1-ISVsa3* and *ISVsa3-orf2-abh-tet(X4)*, respectively. Direct alignment of tet(X) genetic environments in

Flavobacteriaceae with that of *Acinetobacter* spp. and Enterobacteriaceae returned no hits. However, a 397-bp fragment composed of partial fragments of the mobile element *ISVsa3* and the *virD2* gene, which are uniquely detectable in members of *Acinetobacter* spp. and Enterobacteriaceae, was identified in the genome of two *C. bernardetii* strains.”.

Q5: Ln 161 – Two *Acinetobacter* were positive for tet(X) but negative for a phenotype. Were the genes expressed (a simply RT-PCR experiment could address this question)?

Response: We have performed qPCR of *tet(X)* in the *Acinetobacter* strains and added the results in both methods and results sections.

Q6: Ln 207. This reviewer concurs with the conclusion about ancestry, although the authors should point out in the discussion that there could be a non-Flavobacteriaceae ancestor that passed the ancestral tet(X) gene to this family and Proteobacteria (i.e., polyphyletic events).

Response: The following sentence was added in the discussion “Despite these observations, we cannot rule out the possibility that there was a non-Flavobacteriaceae ancestor which passed the ancestral tet(X) gene to Flavobacteriaceae and Proteobacteria on occasions such as polyphyletic events.”.

Q7: Figure 4 – authors need to label the clades.

Response: The clades have been labeled.

Q8: L305 – as with the introduction, the authors attribute the emergence of tigeicycline-resistant bacteria to “abusive use of antibiotics.” They made no attempt to quantify how antibiotics were being used. In fact, “any” use of tetracycline antibiotics has the potential to select for such resistance, whether anyone considers this abusive or not. Best to avoid value judgements in the absence of data.

Response: Thanks for the comment. We have removed the word “abusive” from this sentence.

Q9: L311 – the authors argue that the “molecular mechanisms underlying dissemination of these genes in clinical settings” is poorly understood...and presumably, this is the rationale for their study (next sentence). Their study, however, has nothing to do with dissemination in clinical settings. They are able to provide a prevalence value based on a taxa, and they make a reasonable argument that Flavobacteriaceae is the ancestral source of tet(X) genes. They were unable to demonstrate horizontal gene exchange, and the fact that none of the non-Flavobacteriaceae examined had a tet(X2) orthologue suggests that horizontal gene exchange, if it occurs, is infrequent. In this case, dissemination in clinical settings is really an issue of infectious organisms colonizing new human hosts – the language should reflect greater precision of meaning.

Response: We agree that this study has nothing to do with dissemination of tet(X) in clinical

settings. We have rephrased the original sentence “molecular mechanisms underlying dissemination of these genes in clinical settings remain poorly investigated” to reflect the current situation. The sentence has been revised as “In clinical settings, dissemination of tet(X) orthologues clonally or via horizontal gene transfer could result in extensive colonization of tetracycline resistant organisms in human”.

Q10: L263-266 – the authors speculate that genetic exchange occurred sometime in the past. This argument is consistent with the limited data available (previous sentence). I recommend that the authors point out that this did not have to occur by conjugation. Some of the bacteria in question (e.g., *Acinetobacter*) can be naturally competent and may have scavenged the DNA containing *tet(X)* in the environment (both are considered environmental bacteria). It would be useful if the authors could estimate when such an exchange might have occurred (millions of years ago?).

Response: The following sentence was added to point out that this genetic exchange did not have to occur by conjugation “Of note, such genetic exchange events could occur in processes other than conjugation. For example, some of the bacteria (e.g., *Acinetobacter* sp.) can be naturally competent and may have scavenged DNA fragments containing tet(X) from the environment”.

We appreciate your comment on estimating the time when the genetic exchange might occur, but due to the lack of sufficient evidences (the genetic information obtained in the current study is not sufficient to infer a date), we are not able to include such information in the manuscript.

Q11: The supplement, as provided to the reviewer, was unreadable (excel page spread across multiple pdf pages), but presumably this is where the reader can figure out which strain codes apply to which strains. In places where the strains can be given more complete descriptions (e.g., genus and species), it is best to provide this information for the convenience of your reader (e.g., Figure 5).

Response: We apologize for this inconvenience. The supplement is an excel file containing strain information. We have to keep it with its original format. In this case, we expanded the legend of Figure 5 and included more information on the strains (genus and species). The following sentence was added to the Legend of Figure 5 “ICE sequences aligned in this figure, from top to bottom, were from tet(X)-positive strains *Chryseobacterium indologenes* CI6, *Chryseobacterium bernardetii* CB1, *Chryseobacterium* sp. C3, *Elizabethkingia anophelis* EA1, *Elizabethkingia anophelis* EA3, *Elizabethkingia meningoseptica* EM1 and *Chryseobacterium lactis* CL2 respectively.”.

Also, In legend of Figure 2, the following description was added “Labels in the Y axis represent the species of the strain, AB, *Acinetobacter baumannii*; AN, *Acinetobacter nosomialis*; C, *Chryseobacterium* sp.; CB, *Chryseobacterium bernardetii*; CF, *Citrobacter freundii*; CI, *Chryseobacterium indologenes*; CL, *Chryseobacterium lactis*; EA, *Elizabethkingia anophelis*; EC, *Escherichia coli*; EF, *Empedobacter falsenii*; EM,

Elizabethkingia meningoseptica; PX, *Pseudomonas xiamenensis*; SM, *Sphingobacterium mizutaii*.”

Q12: Figure 1A is labeled as describing the “prevalence” but it actually describes the “distribution”

Response: We have changed “prevalence” to “distribution” in the caption of Figure 1A.

Q13: Ln 383 – authors need to point out that these are all clinical isolates in the methods – and identify their human subjects clearance (or reason for exemption).

Response: We added the sentence “The study comprised two phases, both of which involved analysis of clinical isolates collected from hospitals in China” to point out the fact that all isolates are from clinical settings. Human subject clearance was included in the manuscript as follows “Ethical permission for this study was approved by the Zhejiang University ethics committee (reference No. 2019-074).”.

Q14: Ln463 – whole genome sequences are deposited in GenBank, but it looks like there were other 16S sequences that also need to be deposited, correct?

Response: We have deposited the 16S rRNA sequences in the NCBI database. The following sentence was added in the manuscript “The 16S rRNA gene sequences of all strains in this study have been deposited in the NCBI GenBank database under accession numbers MT793124-MT793229”

Reviewer #2 (Remarks to the Author):

I would like to thank the authors for their manuscript. Resistance to novel antibiotics such as tigecycline is an important topic and the molecular epidemiology of the tet(X) this study adds to this area. However, I have questions, suggestions and thoughts to add to the manuscript. Please refer to the points below for a complete review:

Abstract

Q1. Line 30: replace report with resort

Response: amended accordingly.

Introduction:

Q2. Line 104: The authors could add specifically that they are looking at multiple genera of

clinical isolates specifically as there seems to be a knowledge gap in this area.

Response: We have revised according to the reviewer's comment. Now the sentence reads "In this study, we conducted a nationwide surveillance and genetic characterization of multiple genera of clinical isolates collected during the period 1994 to 2019 in China to fill this knowledge gap."

Results

Q3. Line 110-113: While the sample size is large and covered many hospitals in a large area, I wanted to confirm with the authors if they can assume that their study population can be representative of China. Were samples collected as part of an active surveillance process? If they could elaborate a little more on this in the manuscript (either in methods or in their results section), that would be helpful.

Response: The study population is representative of China since all isolates were acquired from active surveillance. We added the following sentence in the Methods "All isolates were collected as part of an active surveillance process conducted in China".

Q4. Line 122: Replace Burkholderi with Burkholderia. Please check the manuscript for additional changes to Burkholderia spp.

Response: amended accordingly.

Q5. Line 196: Please provide the reference for the study that looked at tet(X) from animal sources that was used to create Figure 3.

Response: The reference is provided. According to the journal policy, at most 70 references can be cited in the main text. We thus included other references used to create Figure 3 in supplementary Table S2.

Q6. Figure 3: "Text with red fonts depicts tet(X)-positive isolates". Please expand, do you mean these isolates are those that were tested in this current study?

Response: The sentence has been revised as "Text in red fonts depicts tet(X)-positive isolates tested in this current study, and information of isolates with black fonts were retrieved from the literature (Table S2).".

Q7. Lines 223-225: "The first clade of Tet(X) protein evolved among Flavobacteriia and Sphingobacteria and did not spread to other species. All Tet(X) proteins in this clade remained in the Tet(X) and Tet(X2) family". It would be helpful if the authors could add which isolate had which Tet(X) gene variant in Figure 4. Just by looking at Figure 4, I am unable to tell that the isolates in the first clade had Tet(X) and Tet(X2).

Response: We carefully checked the sequences in Clade I and found they should not be

assigned to the published tet(X) orthologues because their amino acid sequence only exhibited 60%-64% similarity with that of Tet(X). We thus did not label the tet(X) variant in Clade I of Figure 4. The original sentence “All Tet(X) proteins in this clade remained in the Tet(X) and Tet(X2) family” was replaced by a new sentence “Tet(X) candidate proteins in this clade were 60%~64% identical to that of Tet(X) in terms of amino acid sequence”.

Q8. Figure 4: I do not understand the coding scheme of Figure 4. The authors mention “Orange, blue, yellow and green backgrounds denote bacteria belonging to Bacteroidia, Flavobacteria, Gammaproteobacteria and Sphingobacteria respectively”. However, the backgrounds are discrepant. For instance, the authors have highlighted Salmonella Stanley (which should be classified as Gammaproteobacteria) as blue which indicates that it is belonging to Flavobacteria. Unless I am mistaken about the purpose of the blue/green highlights in the figure, please cross-check and/or re-write.

Response: Sorry for the oversight. We just used two background colors to indicate the phylum of the bacteria, so the original sentence has been revised as follows “Blue and green backgrounds denote bacteria that belong to Proteobacteria and Bacteroidetes, respectively.”.

Discussion:

Q9. The authors could also include the importance of finding tet(X) genes in both clinical and human isolates and what it means for human health due to horizontal gene transfer.

Response: The following sentence was included in the discussion “In clinical settings, dissemination of *tet(X)* orthologues clonally or via horizontal gene transfer could result in extensive colonization of tetracycline resistant organisms in human.”.

Q10. Another important discussion point that the authors should consider adding in their discussion is while they observed high levels of tigecycline resistance, few isolates carried tet(X) genes. They should consider highlighting other genes that are responsible for conferring tigecycline resistance.

Response: The following paragraph was added to the discussion section “According to data obtained in our surveillance, 11.17% of the test isolates exhibited tigecycline resistance, yet only 1.58% of all isolates carried tet(X)-like genes, suggesting that factors other than Tet(X) mediated the majority of tigecycline resistance in clinical settings. The over-expression of chromosomal efflux pumps of the RND family and those encoded by the plasmid-borne *tmxCD1-toprJ1* gene cluster, mutations in the ribosomal binding site such as the *rpsJ* gene, and mutations in the plasmid-mediated efflux pump genes *tet(A)* and *tet(L)* were reported to be associated with tigecycline resistance phenotypes. Factors that confer the drug susceptibility phenotypes of tigecycline-resistant bacteria remained to be investigated.”.

Q11. The authors should also consider why the tet(X) gene may be important for other tetracycline antibiotics, especially due to their widespread use in veterinary medicine.

Response: The following sentence was added in the discussion “The selective pressure imposed by the continuous application of tetracyclines in veterinary medicine could serve to maintain and spread the tet(X) genes among pathogenic microorganisms”.

Q12: Overall thoughts:

The manuscript is well-written and has important implications on the molecular epidemiology, evolution and transmissibility of the tet(X) gene. The authors articulated clearly why tigeicycline resistance is a major problem and highlighted the missing gaps in information related to surveillance of the tet(X) gene in China. I also appreciate the inclusion of already existing tigeicycline resistance surveillance system in the introduction section which gives the readers a good idea about what data is being collected and what is lacking. There were a few points in the results section where I was a little unclear about the results. If the authors could address these concerns, this would further strengthen the results. While going through Table 1, I did notice that there may be phenotypic and genotypic discrepancies between phenotypic tigeicycline resistance and presence/absence of tet(X) gene. The authors should consider this point as well. The Materials and Methods section had information to ensure reproducibility of the work. I would recommend the authors to consider adding additional discussion points in their manuscript highlighted above to strengthen their conclusions.

Response: Thank you for the comments. The tigeicycline resistance rates and presence/absence of the tet(X) gene was included in the results. Furthermore, we added discussion on the underlying reasons for the discrepancies (see response to comment 10).